# STABILIZING ADVERSARIAL INVARIANCE INDUCTION BY DISCRIMINATOR MATCHING

## ABSTRACT

Incorporating the desired invariance into representation learning is a key challenge in many situations, e.g., for domain generalization and privacy/fairness constraints. An adversarial invariance induction (AII) shows its power on this purpose, which maximizes the proxy of the conditional entropy between representations and attributes by adversarial training between an attribute discriminator and feature extractor (Xie et al., 2017). However, the practical behavior of AII is still unclear as the previous analysis assumes the optimality of the attribute classifier, which is rarely held in practice. This paper first analyzes the practical behavior of AII, indicating that AII has theoretical difficulty as it maximizes variational *upper* bound of the actual conditional entropy, and AII catastrophically fails to induce invariance even in simple cases as suggested by the above theoretical findings. We then argue that a simple modification to AII can significantly stabilize the adversarial induction framework and achieve better invariant representations. Our modification is based on the property of conditional entropy; it is maximized if and only if the divergence between all pairs of marginal distributions over representations between different attributes is minimized. The proposed method, *invariance induction by discriminator matching*, modify AII objective to explicitly consider the divergence minimization requirements. Empirical validations on both the toy dataset and four real-world datasets (related to applications of user anonymization and domain generalization) reveal that the proposed method provides superior performance when inducing invariance for nuisance factors.

## 1 INTRODUCTION

Extensive studies have demonstrated that deep neural networks (DNNs) can uncover complicated variations in data to provide powerful representations that are useful for classification tasks (Hinton et al., 2006; Krizhevsky et al., 2012). However, in some scenarios, the learned representation should be invariant to some attribute of the input data. A motivating example is a task called domain generalization (Blanchard et al., 2011), which requires learning a domain-invariant representation that applies to unseen domains (e.g., the data of an unseen user or different image sources). When practitioners apply DNNs to data that include a large amount of user information (such as images with usernames (Edwards & Storkey, 2016) or data from wearables (Iwasawa et al., 2017)), the desired representations should not include user identifying information. For legal and ethical reasons, machine learning algorithms must make fair decisions that are independent of sensitive variables such as gender, age, or race (Louizos et al., 2016). Therefore, this study aims to answer the following question: how can we systematically incorporate the desired invariance into representation learning?

Invariance induction is a systematic solution to this problem, which often introduces an additional regularization term that measures the level of invariance (Edwards & Storkey, 2016; Iwasawa et al., 2017; Xie et al., 2017). One theoretically sound metric is the conditional entropy of attributes given representations of data $H(a|z) = \mathbb{E}_{p(z,a)}\left[-\log p(a|z)\right]$, where $a$ and $z$ denote the random variables of attributes and representations, as it is maximized if and only if the representations are invariant to attributes. However, exact calculation is intractable as $p(a|z)$ is unknown in general.

Recently, Xie et al. (2017) introduced a sensible way to approximate the conditional entropy (Figure 1-(a)), which uses probabilistic attribute classifier $q_\phi$, where $\phi$ represents the parameter of the classifier, to approximate the conditional probability distribution $p(a|z)$. One can estimate the desired conditional entropy by simply substituting $p(a|z)$ to the approximated conditional distribution

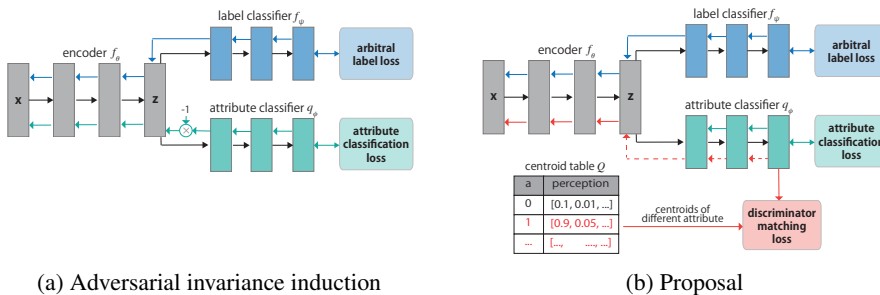

(a) Adversarial invariance induction        (b) Proposal

Figure 1: Comparison between the previous method and the proposed method. As with AII, the proposed method ensure the invariance to the nuisance attribute by deceiving the external attribute classifier, but the proposed method deceives it deferentially. The detail is explained in Section 4.

$q_\phi(a|z)$. The estimated conditional entropy is then used to update the weights of the encoder $f_\theta$ so that the updated representations increase the approximated conditional entropy. Since $f_\theta$ and $q_\phi$ is an adversarial relationship, this framework is called *adversarial invariance induction (AII)*. A similar approach was extensively used in domain generalization, fair-prediction, and privacy-protection contexts (Edwards & Storkey, 2016; Motiian et al., 2017; Xie et al., 2017; Iwasawa et al., 2017).

Build upon the above achievements, this paper examines a way to improve the stability of the adversarial invariance induction framework. Note that, Xie et al. (2017) already demonstrated that the above procedure (alternatively optimizing attribute classifier and feature extractor) possesses an equilibrium where the encoder maximizes the true conditional entropy. However, their analysis impractically assumes $q_\phi(a|z) = p(a|z)$, so little is known about the practical behavior of AII and how to improve it. Indeed, several prior works report unsuccessful results of AII (Xie et al., 2017; Moyer et al., 2018). To further examine the practical behavior of AII, we first analyze the behavior of AII from the lens of the variational approximation. Our analysis shows that AII maximizes the variational *upper* bound of the conditional entropy $H(a|z)$, and AII objective itself is not upper-bounded. This result suggests that the optimization of AII, without the assumption of the optimality of $q_\phi(a|z)$, does not need to maximize the conditional entropy, as the maximizing upper bound does not give any guarantees. Empirical validations show that AII catastrophically fails to induce invariance even in satisfactory simple situations, as suggested by the above finding.

We then argue that the simple modification to AII attains better property from the optimization perspective while achieving the same goal asymptotically. Our modification is based on the property of the conditional entropy, i.e., it is maximized if and only if the divergence between all pairs of marginal distributions over $z$ between different attributes are minimized (under the uniform assumption of $p(a)$). This property suggests that the invariance induction algorithm should also minimize the divergence, which is not considered on AII and possibly induce unstable behavior. In contrast, our proposed method, *invariance induction by discriminator matching (IIDM)*, explicitly considers the divergence minimization requirements by minimizing the proxy of the divergence between the marginals, which push the representations with different attributes are recognized similarly by the discriminator (as shown in Figure 1-(b)). We discuss the relationship among our proxy, the divergence between the marginals, and non-saturating heuristics used in different but related community (Goodfellow et al., 2014).

The main contributions of this paper can be summarized as follows. (1) We highlight the practical issues of AII, which is a state-of-the-art framework for invariance induction. (2) We propose a modification to AII by explicitly consider the property of the maximum conditional entropy. Empirical validations on the toy dataset and four real-world datasets demonstrate that the proposed method provides superior performance and faster convergence at the level of invariance induction. For example, in the experiments on toy dataset, the proposed method converges to mostly the maximum conditional entropy, while the optimization of AII is catastrophically unstable. Gif visualizations available at `https://drive.google.com/open?id=1N2kuTCfwjQBGgv3dxTSD9oBklAenLj-J` (see Section 4.3 for more detail).

## 2 RELATED WORKS

The goal of the invariance induction is to learn attribute-invariant spaces given a training dataset made of tuples of $\{(x_n, y_n, a_n)\}_{n=1}^N$ (supervised setting) or pairs of $\{(x_n, y_n)\}_{n=1}^N$ (referred to as

unsupervised setting (Jaiswal et al., 2018)), where $x_n$ is an observation and $y_n$ is a target of $x_n$, $a_n \in \mathcal{A}$ is a realization of a categorical random variable $a$. This setting arises in many application areas. For example, Iwasawa et al. (2017) considers the privacy-preserving activity recognition problem using the data of wearables, where one wants to learn user invariant representations while keeping the information about activity into the representations. In this case, the attribute $a$ corresponds to some sensitive user information, and $y$ is the activity label. In the context of fairness-aware decision making, one needs to ensure the algorithm to make fair decisions, which often done by statistical away the sensitive attributes $a$ (such as gender, age, or race) from the representations (Edwards & Storkey, 2016). To build a classifier robust to the domain shifts (called domain generalization Blanchard et al. (2011)), learning representations invariant to domain shifts is a popular approach (the domain label is the attribute in this case) (Muandet et al., 2013; Ghifary et al., 2015).

Assume $f_\theta$ is a function (encoder) that parameterized by a neural network, which maps observations $x$ to representations $z \in \mathcal{Z}$. Typically, the invariance induction algorithm use differentiable dependency measurements and use it as a regularization term to learn $f_\theta$: $\mathbb{E}\left[L(f_\theta(x_n), y_n) + \lambda V(f_\theta(x_n), a_n)\right]$, where a $\lambda$ is the weighting parameter, $V$ somehow measures invariance of the representations regarding the attribute, and $L$ is a loss function that represents how much information about $y$ is present in the representations.

*Adversarial invariance induction* (AII) is a recently proposed approach for measuring the $V$ by an external neural network. That is, if the external network can accurately predict $a$ from $z = f_\theta(x)$, AII considers $z$ to have considerable information about $a$. The external neural network is often called as a discriminator or adversary in this context. Information from the discriminator is used to update the weights of the encoder $f_\theta$ so that the updated representations have less information about $a$. Mathematically, the method optimizes following min–max game:

$$\min_{\theta,\psi} \max_{\phi} \mathbb{E}_{p(x,a)}[-\log q_\psi(y_n|z_n{=}f_\theta(x_n)) + \lambda \log q_\phi(a_n|z_n{=}f_\theta(x_n))], \tag{1}$$

where $q_\psi(y|.)$ and $q_\phi(a|.)$ is a conditional probability distribution approximated by a categorical classifier parameterized by $\psi$ (for $y$) and $\phi$ (for $a$) respectively. By alternatingly (or jointly with the gradient reversal layer (Gan et al., 2016)) optimizing $\theta$ and $\phi$, this framework ensures that there is little or no information about the nuisance attributes in the representations.

As Xie et al. (2017) demonstrated, the above min–max game can be regarded as a way to estimate the conditional entropy $H(a|z)$ and use the estimates to induce the invariance. The key advantage of AII is that this framework does not depend on the pre-defined metrics such as L2 distance and maximum mean discrepancy (Zemel et al., 2013; Li et al., 2014; Louizos et al., 2016)), which is often easy to compute yet less powerful. Formally, Xie et al. (2017) demonstrated that the above adversarial game has an equilibrium where the encoder maximizes the true conditional entropy, under the assumption that $q_\phi(a|z)$ correctly estimates the true $p(a|z)$. However, their analysis assumes the optimality of $q_\phi(a|z)$, which is rarely held in practice. This paper analyzes the behavior of AII without such an assumption to examine the practical behavior of AII.

It is noteworthy that the above formulation resembles the original formulation of GAN (Goodfellow et al., 2014) and domain adversarial networks (DAN) (Ajakan et al., 2014; Gan et al., 2016), which is never used practically. Instead, they often incorporate several heuristics, e.g., non-saturating heuristics in GAN, and asymmetric mapping in domain adaptation (Tzeng et al., 2017). These works motivate us to replace the min–max game of the adversarial invariance induction problem.

Although no prior works in the invariance induction community have been explicitly considered yet, one can transfer the non-saturating heuristic used in the GAN via label flipping:

$$\min_{\theta} \mathbb{E}_{p_\theta(z,a)}\left[-\sum_{a_j \neq a_n} \log q_\phi(a_j|z_n = f_\theta(x_n))\right], \tag{2}$$

which also enhances the discriminator to misclassify the attribute of the data. In the remainder of this paper, we refer to this version as the *non-saturating version* and denote it by NS. As discussed in the GAN community, this objective provides meaningful gradients because it not saturated even if the discriminator is a supremum. Later, we discuss the connection between the proposal and this objective, and how our proposal attains better properties while inheriting the non-saturating nature of the NS objective.

# 3 PRACTICAL BEHAVIOR OF ADVERSARIAL INVARIANCE INDUCTION

In this section, we analyze the practical behavior of AII framework. As mentioned above, the prior analysis of AII focus on the ideal situation where one can access the perfect attribute classifier, i.e., $q_\phi(a|z) = p(a|z)$. What happen if the optimality does not holds? As the optimally rarely holds in practice, it is important to analyze the behavior under the absence of the optimality.

## 3.1 VARIATIONAL BOUNDS

To examine the practical behavior of AII, we first derive the general relationships between the conditional entropy and AII framework, without assuming $q_\phi(a|z) = p(a|z)$. In the remainder of this paper, we assume the use of alternating optimization to solve the adversarial game. At each iteration, AII first updates the attribute classifier $\kappa$ times by maximizing eq. 1 while fixing $f_\theta$. AII then updates the encoder by minimizing the eq.1 while fixing $q_\phi$.

The conditional entropy and AII objective has the following relationships:

$$
\begin{aligned}
H(a|z) = \mathbb{E}_{p_\theta(z,a)}\left[-\log p(a_n|z_n)\right] &= \mathbb{E}_{p_\theta(z_n,a_n)}\left[-\log q_\phi(a_n|z_n)\right] - D_{KL}(p(a|z)||q_\phi(a|z)) \quad (3) \\
&\leq \mathbb{E}_{p_\theta(z_n,a_n)}\left[-\log q_\phi(a_n|z_n)\right] = H_q(a|z), \quad (4)
\end{aligned}
$$

where $D_{KL}$ represents the Kullback–Leibler (KL) divergence, which is greater than zero by definition, and $p_\theta(z)$ is an empirical distribution of representations. Maximizing $H_q(a|z)$ yields the encoder update in AII framework exactly. The update of the discriminator is equivalent to minimizing this KL divergence. Therefore, the update of the discriminator tightens the bound. The bound is tight when $D_{KL}(p(a|z)||q_\phi(a|z)) = 0$. In this case, maximizing $H_q(a|z)$ is equivalent to maximizing the true conditional entropy, recovering the previous analysis (Xie et al., 2017).

Unfortunately, it is not practical to assume the optimality of $q_\phi(a|z)$ due to the limitation of the capacity of the $\phi$ and computational costs to increase $\kappa$. In such a practical situation, AII maximizes the *upper* bounds of the conditional entropy, as shown in the eq. 4. This challenge AII as maximizing the upper bound of the function of interest $f$ does not guarantee the maximizing the $f$ in general (in our case, $f$ is the conditional entropy). For example, in the most exaggerated case, moving the entire distribution in a specific direction increases the upper bound while the actual conditional entropy remains constant. The above observations do not immediately mean that maximizing $H_q(a|z)$ (which is done in AII) is useless. However, it implies that there are regions where maximizing $H_q(a|z)$ does not increase $H(a|z)$.

## 3.2 DIVERGENCE MINIMIZATION REQUIREMENT

It is noteworthy that $H_q(a|z)$ itself is not upper-bounded, as assigning zero probability for correct pair of $z, a$ makes $H_q(a|z)$ infinity. As the true conditional entropy has maximum values by definition, it suggests that AII's optimization has an incentive to update the parameter of the encoder even when the conditional entropy is successfully maximized. Why AII has such an incentive? Here, we derive *divergence minimization requirements* for maximizing the conditional entropy and how it challenges AII optimization.

Specifically, the maximum conditional entropy holds the following property:

**Proposition 1.** *We assume that $A$ is a uniform categorical random variable. The maximum conditional entropy $H(a|z)$ is $-\log \frac{1}{K}$, and $H(a|z)$ is maximized if and only if $p(z|a_i) = p(z|a_j)$ for all $a_i \neq a_j \in \mathcal{A}$ and $z \in \mathcal{Z}$.*

The proof is shown in the Appendix A. This proposition means that maximizing conditional entropy is asymptotically equal to minimizing pairwise divergence. Assume $D$ is a divergence measurement over a space of a possible probability distribution (i.e., $D(p||q) \geq 0$ for all $p, q$ and $D(p||q) = 0$ if and only if $p = q$). For simplicity, we denote the marginal distribution over a random variable $z$ associated with an attribute $a = a_i$ as $p_\theta^i(z)$. Then, the following corollary holds.

**Corollary 1.** *If $f_\theta$ gives an attribute invariant representation (i.e., conditional entropy $H(a|z)$ is maximized), then $D(p_\theta^i(z)||p_\theta^j(z)) = 0$ for all $a_i, a_j$ and vice versa.*

The above analysis suggests that the invariance induction algorithm should also minimize the divergence between marginal distributions of different attributes $a$. However, the original AII does not

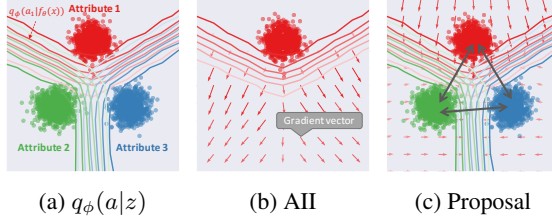

(a) $q_\phi(a|z)$       (b) AII       (c) Proposal

Figure 2: Visualizing how AII and the proposed method update the encoder. The arrow in (b, c) represents the gradient vectors. (a) Both methods utilize $q_\phi(a|z)$, which represented as counterplot in the figure. (b) AII update $f_\theta$ by keeping a distribution away from the decision boundary without considering the information $p_\theta^{i\neq j}(z)$. (c) Our proposal considers both the decision boundary and the information from the marginal distribution of different attribute $p_\theta^{i\neq j}(z)$.

consider such constraints as it only keeps the distribution $p_\theta^i(z)$ away from the non-desired point where a discriminator correctly predicts the attribute. Figure 2-(a, b) visualize how AII updates the feature extractor, assuming there is three gaussian distribution where each distribution corresponds to different attributes. The arrow in the figure represents the direction of the gradient when updating the $f_\theta$ (updating the distribution $p_\theta(z)$). The gradient vector suggests that AII has an incentive to move the distribution regardless of whether it aligns marginal distributions of different attributes.

## 4   INVARIANCE INDUCTION BY DISCRIMINATOR MATCHING

### 4.1   DISCRIMINATOR MATCHING

Based on the analysis, we propose simple modifications to AII considering the divergence minimization requirements. One possible approach to induce this divergence minimization requirement is to cast the representations on some distribution whose divergence is easy to compute. For example, KL divergence is easy to compute if both marginal distributions are Gaussian. However, casting the distribution harms the representative power of the deep encoder as selecting the proper prior is difficult in general. Moreover, if the representations have much information about the nuisance attribute and, therefore, the marginal distributions between different attributes do not have supports in the representation spaces, $\mathcal{D}(p||q)$ generally does not exists.

We propose a simple alternative by defining the proxy of the divergence using the learned critic function $q_\phi$. Specifically, we consider a divergence over $q_\phi^i(a)$, where $q_\phi^i(a) = \int p_\theta^i(z)q_\phi(a|z)dz$. In the remainder of the paper, we assume to use KL-divergence as divergence measurements, though it could be a design choice in practice. IIDM minimizes the following discriminator matching loss for all pairs of $a_i$ and $a_j \neq a_i$:

$$V_{dm}(p_\theta^i(z)||p_\theta^j(z); \phi) := \mathbb{E}_{z_j \sim p_\theta^j(z)}\left[D_{KL}(q_\phi^i(a)||q_\phi(a|z_j))\right]. \tag{5}$$

Intuitively speaking, IIDM restricts the update of the encoder to consider the location of the marginal distributions of different attributes, in addition to the decision boundary (which is all the information source of the AII). In other words, IIDM alleviates the possibility that the encoder moves the distribution regardless of whether it aligns marginal distributions of different attributes. Note that, such a divergence minimization perspective is missing in the original AII, as mentioned in Section 3.2. For reference, Figure 2 compares how AII and the proposed method update the feature extractor.

Formally, the discriminator matching objective $V_{dm}$ is related to the divergence $D_{KL}(p_\theta^i(z)||p_\theta^j(z))$ through the divergence between $D_{KL}(q_\phi^i(a)||q_\phi^j(a))$. Firstly, we can derive

$$D_{KL}(q_\phi^i(a)||q_\phi^j(a)) \leq \mathbb{E}_{z_j \sim p_\theta^j(z)}\left[D_{KL}(q_\phi^i(a)||q_\phi(a|z_j))\right], \tag{6}$$

by using the Jensen's inequality. Also, based on the data processing inequality of the f-divergence (Gerchinovitz et al., 2017; Barber et al., 2018), the following inequality holds:

$$D_{KL}(p_\theta^i(z)||p_\theta^j(z)) \geq D_{KL}(q_\phi^i(a)||q_\phi^j(a)). \tag{7}$$

See the Appendix B for more detail.

---

**Algorithm 1** Pseudo code of IIDM

---

**Require:** Initialize weights of neural networks $\{\theta, \phi, \psi\}$, and the centroid $Q_{(0)}$ with all data $\mathcal{S}$.
  **while** training() **do**
    **while** repeat $\kappa$ times **do**
      update the attribute classifier by eq. 9 with the bath of data $\mathcal{B} \sim p(x, y, a)$.
    **end while**
    sample the batch of data $\mathcal{B} \sim p(x, y, a)$.
    split the bath of data $\mathcal{B}$ into $\mathcal{B}_0, \cdots \mathcal{B}_{K-1}$ where $\mathcal{B}_i$ is the batch of data with the attribute $a_i$.
    $Q_t^i \leftarrow \gamma Q_{(t-1)}^i + (1-\gamma)\frac{1}{|\mathcal{B}_i|}\sum_{x_i \in \mathcal{B}_i} q_\phi(a|f_\theta(x_i))(\forall i \in \mathcal{A})$.
    update the feature extractor and label classifier by eq. 10 with the batch of the data $\mathcal{B}$.
  **end while**

---

As $V_{dm}$ is an upper bound of the divergence $D_{KL}(q_\phi^i(a)||q_\phi^j(a))$, minimizing $V_{dm}$ surely minimizes the KL-divergence. Besides, in a special case where the attribute classifier is invertible, $V_{dm}(\theta, \phi) = 0$ ensures that $p_\theta^i(z) = p_\theta^j(z)$, as minimizing $D_{KL}(q_\phi^i(a)||q_\phi^j(a))$ is equivalent to minimizing $D_{KL}(p_\theta^i(z)||p_\theta^j(z))$ (Barber et al., 2018). In other words, minimizing $V_{dm}$ has the same fixed points with conditional entropy maximization if the discriminator is invertible. Restricting the invertibility of the discriminator is an interesting direction. However, we did not add such a regularization as restricting neural networks is difficult in general and open research areas (Jacobsen et al., 2018; Behrmann et al., 2018; Ardizzone et al., 2018). Instead, we empirically validate that the proposed method reliably learns invariant representations even without such a regularization.

It is worth mentioning that the discriminator matching loss is closely related to the non-saturating version of AII. Specifically, the proposed method minimizes

$$\mathbb{E}_{z_j \sim p_\theta^j(z)}\left[D_{KL}(q_\phi^i(a)||q_\phi(a|z_j))\right] = \mathbb{E}_{z_j \sim p_\theta^j(z)}\left[\sum_{a \in \mathcal{A}} -q_\phi^i(a)\log q_\phi(a|z_j))\right] + C,$$

and the NS objective (Eq. 2) can be rewritten as

$$V_{NS}(\theta; \phi) = \sum_{a_j \in \mathcal{A}}\sum_{a_j \neq a_i}\mathbb{E}_{z_j \sim p_\theta^j(z)}\left[\sum_{a \in \mathcal{A}} -p^i(a)\log q_\phi(a|z_j))\right],$$

where $p^i(a)$ is equal to 1.0 for $a = a_i$ and zero otherwise. Ignoring the constant term, the only difference is whether to push $q_\phi(a|z_j)$ to $q_\phi^i(a)$, or true marginal $p^i(a)$. Because the discriminator matching loss considers $q_\phi(a \neq a_j|z_j)$, similar to the NS objective, its gradient does not vanish even if $q_\phi$ is the supremum. This property encourages faster convergence as with the NS loss. It differs from NS as it considers the divergence minimization requirements, while NS only considers the current decision boundary of the discriminator but does not consider information from $p_\theta^i(z)$ directly (similar to AII). Therefore, NS push representations as far as possible as long as it successfully changes the discriminator's prediction regardless of whether it aligns marginal distributions.

### 4.2 ALGORITHM

One implementation choice is how to calculate $q_\phi^i(a)$ in Eq. 4.1. The straightforward approach is through Monte Carlo approximation: $q_\phi^i(a) = \int p_\theta^i(z)q_\phi(a|z)dz = \mathbb{E}_{p_\theta^i(z)}[q_\phi(a|z)]$. Although it is an unbiased estimation, the variance is large if the number of samples is small. The average can be calculated from all samples (or a sufficiently large number of samples from each $K$ attributes) at every iteration. However, it requires additional computation other than standard mini-batch estimation. Moreover, the computation becomes intractable as the number of attribute values grows.

We address these issues by using the moving centroid mechanism. Instead of estimating $q_{\theta,\phi}^i(a)$ every time with sufficiently large samples, the proposed method stores the moving average of $q_\phi^i(a)$:

$$Q_t^i(a) = \gamma Q_{t-1}^i(a) + (1-\gamma)q_t^i(a), \tag{8}$$

where $Q_{t-1}^j$ is a previous centroid, $q_t^i(a)$ is the new estimation of the centroid based on a single batch, and $\gamma$ is the decay parameter for controlling the speed at which the centroids change. We initialized $Q_0^j$ by computing the centroids of all training data points.

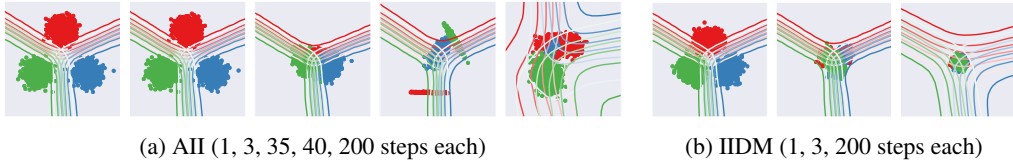

(a) AII (1, 3, 35, 40, 200 steps each)                    (b) IIDM (1, 3, 200 steps each)

Figure 3: Visualizing behaviors of AII and IIDM on toy datasets (`https://drive.google.com/open?id=1N2kuTCfwjQBGgv3dxTSD9oBklAenLj-J` contains gif version).

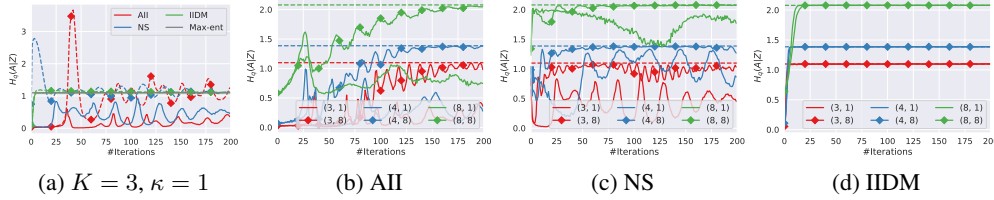

(a) $K = 3, \kappa = 1$      (b) AII      (c) NS      (d) IIDM

Figure 4: Quantitative comparison of AII, NS and IIDM (proposed method) on the toy datasets.

Then we can use the standard mini-batch method to calculate the discriminator matching loss. As with the AII, IIDM incorporates alternating optimization. Specifically, as with AII, IIDM firstly update the attribute classifier $q_\phi$ by

$$\max_\phi \mathbb{E}_{p(x,a)}[\log q_\phi(a_n|z_n{=}f_\theta(x_n))], \tag{9}$$

and update the encoder and the classifier by

$$\min_{\theta,\psi} \mathbb{E}_{p(x,a)}\left[-\log q_\psi(y_n|z_n{=}f_\theta(x_n)) + \lambda\left[\sum_{a_i \neq a_n}[D_{KL}(Q^i(a)||q_\phi(a|z_n{=}f_\theta(x_n)))]\right]\right], \tag{10}$$

where $\psi$ is the parameter of the classifier, the 1st term is a classification loss (which is same with AII), and $\lambda$ is a weighting parameter.

## 4.3 SIMULATION

To clarify the practical issue of AII and the benefit of the proposal, we design the toy dataset, which comprises samples from several Gaussian distributions with different means ($[\sin(\frac{i}{K}\pi), \cos(\frac{i}{K}\pi)]$, and $i \in 1, 2, \cdots K$) and the same variance, assuming that each distribution corresponds to different attributes. We test AII on this synthesized dataset while varying the number of distributions $K$ and $\kappa$ (number of updates of $\phi$ per iteration). Specifically, we first train the discriminator 100 times with a batch size of 128 and update $q_\phi$ and $f_\theta$ iteratively using stochastic gradient descent with a learning rate=0.1. Figure 3 visualizes how distributions move during the optimization of (a) AII and (b) IIDM on synthesized data ($K = 3$ and $\kappa$=1). The Figure 4 represents the quantitative results on this configuration, where solid lines represent the approximated conditional entropy by using a post-hoc classifier $q_{eval}$, which is parameterized by the neural network having the same architecture as that of $q_\phi$. For reference, the theoretical maximum value of the conditional entropy (gray line) and the negative log-likelihood of $q_\phi$ (dashed lines) are also depicted. Figure 4 compares the performances of (b) AII (c) NS, and (d) IIDM on different configurations of $\{K, \kappa\}$, where color denotes different $K$, marker denotes different $\kappa$, and the dashed line denotes optimal values.

We can make the following observations. (1) As shown in the visualization of AII and the vibration of the red lines in Figure 4-a, AII is catastrophically unstable, even such a simple case. Figure 4-a also indicates that, at some points (e.g., around 50 iterations), the approximated conditional entropy becomes significantly large, whereas the true conditional entropy remains constant or even decreases. The issue may be alleviated if the discriminator has a sufficiently large capacity and is trained many times at each iteration, as suggested by a faster convergence of AII when $\kappa = 8$. However, the toy dataset is much simpler than real datasets, and it is fair to say that identifying the supremum is more challenging. (2) NS is superior to AII, but it still unstable near the optimal point, as indicated by the vibration of the plots. (3) IIDM consistently reaches the theoretical maximum values in all configurations, as depicted in Figure 4-d.

### 4.4 REMARKS

**Continuous nuisance attribute:** One may question the applicability of our method to the continuous attribute case, e.g., the goal is to learn age-invariant representations. Firstly, the treatment of continuous R.V. is an open question. The most straightforward yet practically used answer is to discretize the continuous R.V. For example, (Xie et al., 2017) divides an age variable into two groups. We can apply the proposed method with similar discretization.

**Semantic alignments:** One well-known problem of invariant feature learning is determining how to incorporate semantic alignment, i.e., how to align only the pair of samples that have the same semantic (the target label). For this purpose, (Li et al., 2018b) proposes adversarial training based semantic alignment method, which prepares multiple domain classification networks where each classifier specialize for each class $y$. Another merit of the proposed method is that we can enforce semantic alignment with a simple modification, without any additional computational costs. Individually, semantic alignment can be carried out by merely computing the centroids for each (attribute, label) tuple and aligning the $q_\phi(a|z)$ of $\{x, y, a\}$ between only centroids of the same label $y' = y$ but different attributes $a' \neq a$. Since most applications of invariant feature learning require that $L_y$ is also minimized, we also test this modification for all the later-described experiments.

## 5 EXPERIMENTS

### 5.1 EXPERIMENTAL SETTINGS

In addition to the simulation results, we provide experimental results on two tasks (four datasets) relevant to invariant feature learning: (1) user anonymization (Opportunity and USC datasets), and (2) domain generalization (MNISTR and PACS datasets). All experiments were implemented in PyTorch and were run on GPUs (either GTX 1080 or Tesla V100).

For user anonymization tasks, Opportunity and USC datasets were used. This task is to learn anonymized representations ($z$ that does not contain user-identifiable information) while maintaining classification performance. The **Opportunity** dataset (Sagha et al., 2011) consists of sensory data regarding human activity in a breakfast scenario. Each record consists of 113 real-value sensory readings, excluding time information. We considered the task of recognizing 18 classes[1]. Following previous studies (Yang et al., 2015; Iwasawa et al., 2017), we use a sliding window procedure with 30 frames and a 50% overlap. The number of samples was 57,790 in total. We parameterize the encoder using convolutional neural networks (CNN) with three convolution-ReLU-pooling repetitions followed by one fully connected layer and classification by logistic regression, following previous studies (Yang et al., 2015; Iwasawa et al., 2017). The discriminator is a simple feedforward network with 800–400 hidden units. The **USC-HAD** dataset is another activity recognition dataset that consists of 14 subjects (Zhang & Sawchuk, 2012). The data include 12 activity classes[2] that correspond to people's most essential and daily activities. MotionNode, which is a 6 DOF inertial measurement unit, is used to record the output from accelerometers that record six real sensory values. The sliding window procedure, with 30 frames and a 50% overlap, produced 172,169 samples.

The MNISTR and PACS are two typical datasets of domain generalization tasks. The **MNISTR** dataset, derived from MNIST, was introduced by (Ghifary et al., 2015). Its labels comprise the ten digits; domains are created by rotating the images in multiples of 15 degrees: 0, 15, 30, 45, 60, and 75. The domains are labeled with the angle by which they are rotated, e.g., M15 and M30. Each image is cropped to $16 \times 16$ pixel in accordance with a previous study (Ghifary et al., 2015)[3]. Similar to (Ghifary et al., 2015), we used two convolution layers with 32 and 48 filters of $5 \times 5$ kernels, followed by a max-pooling layer and two fully connected layers with 100 hidden units. A discriminator with 100 hidden units is connected to the output of the first fully connected layer. The **PACS** dataset is a relatively new benchmark dataset designed for cross-domain recognition (Li et al., 2017). The dataset has 9991 images in total across seven categories (dog, elephant, giraffe, guitar, house, horse, and person) and four domains of different stylistic depictions (photo, painting,

---

[1] open door 1, open door 2, close door 1, close door 2, open fridge, close fridge, open dishwasher, close dishwasher, open drawer 1, close drawer 1, open drawer 2, close drawer 2, open drawer 3, close drawer 3, clean table, drink from cup, toggle switch, and null

[2] walking forward, walking left, walking right, walking upstairs, walking downstairs, running forward, jumping, sitting, standing, sleeping, elevator up, and elevator down

[3] Specifically, we used the dataset distributed at `https://github.com/ghif/mtae`.

Table 1: Performance comparison of user anonymization tasks. The value is the lowest user-classification accuracy with specific performance degradation $(0.01, 0.03$ points) from CNN.

| dataset | Opp-S1 | | Opp-S2 | | Opp-S3 | | Opp-S4 | | USC | |
|---|---|---|---|---|---|---|---|---|---|---|
| threshold | 0.01 | 0.03 | 0.01 | 0.03 | 0.01 | 0.03 | 0.01 | 0.03 | 0.01 | 0.03 |
| CNN | 0.939 | 0.939 | 0.973 | 0.973 | 0.984 | 0.967 | 0.983 | 0.983 | 0.683 | 0.683 |
| AII | 0.631 | 0.517 | 0.590 | 0.590 | 0.694 | 0.659 | 0.589 | 0.586 | 0.512 | **_0.179_** |
| AII+GP | 0.619 | 0.619 | 0.521 | 0.521 | **0.471** | **0.471** | 0.673 | 0.510 | 0.580 | 0.569 |
| NS | 0.635 | 0.452 | 0.614 | 0.523 | 0.484 | 0.484 | **0.499** | **0.482** | None | None |
| IIDM | **_0.462_** | **_0.417_** | **_0.415_** | **_0.415_** | **_0.409_** | **_0.409_** | **_0.486_** | 0.486 | **0.499** | 0.499 |
| IIDM+ | **0.502** | **0.433** | **0.474** | **0.474** | 0.495 | 0.495 | 0.631 | **_0.461_** | **_0.478_** | **0.478** |

cartoon, and sketch). The diverse depiction styles provide a significant domain gap. We use the ImageNet pre-trained AlexNet CNN (Krizhevsky et al., 2012) as a base network, following previous studies(Li et al., 2017; 2018a). A discriminator with 1024 hidden units is connected to the output of the last fully connected layer.

**Baselines:** To demonstrate the efficacy of the proposed method, we compared it with the following methods. (1) A **CNN** trained on the aggregation of data from all source domains. Although there are special treatments for domain generalization, (Li et al., 2017) reports that CNN outperforms many domain generalization methods on the PACS dataset. (2) **AII** (Xie et al., 2017), is a main baseline. (3) **AII+GP** uses a variant of AII with an additional gradient penalty regularization used in GAN (Mescheder et al., 2018). (4) **RevGrad** is a slightly modified version of AII, which uses the gradient reversal layer (Ganin et al., 2016) to train all the networks (encoder, classifier, and discriminator) at the same time. (5) **NS** is a non-saturating version of AII introduced in section 3 of this paper. (6) **CrossGrad** (Shankar et al., 2018) is regarded as a state-of-the-art method in domain generalization tasks. Note that it does not intend to learn invariant representation, so we use CrossGrad only for comparing domain generalization performance. (7) **IIDM** is our proposal. We used the gradient penalty as well. We also tested semantic alignment version and denoted it as IIDM+.

**Optimization:** For all datasets and methods, we used RMSprop for optimization. For all datasets except PACS, we set the learning rate to $0.001$ and the batch size to $128$. For PACS, we set the learning rate to $5e-5$ and the batch size to $64$. The number of iterations was $10k$, $5k$, $20k$, $30k$, and $50k$ for MNISTR, PACS, Opp, and USC, respectively. For a fair comparison, hyperparameters were tuned on a validation set for each baseline. For the adversarial-training-based method, we optimized weighting parameter $\lambda$ from $\{0.001, 0.01, 0.1, 1.0\}$, except for MNISTR, for which it was optimized from $\{0.01, 0.1, 1.0, 10.0\}$. The value of $\alpha$ for CrossGrad was selected from $\{0.1, 0.25, 0.5, 0.75, 0.9\}$. Unless mentioned otherwise, we set the decay rate $\gamma$ to $0.7$.

**Evaluation:** In all the experiments, we selected the data of one or several domains for the test set and used the data of a disjoint domain as the training/validation data. Accurately, we split the data of the disjoint domain into groupings of $80\%$ and $20\%$. We denote the test domain by a suffix (e.g., MNISTR-M0 denotes that the model is trained with the data from M15, M30, M45, M60, and M75 and evaluated on M0). We conducted 20 validations during training at equal intervals. In each validation, we measured the label classification accuracy (Y-acc) and the level of invariance. We empirically measured the level of invariance by training a post-hoc classifier $D_{eval}$ that tries to predict $a$ over learned representations, following previous studies (Xie et al., 2017; Iwasawa et al., 2017). Specifically, we trained the classifier with 800 hidden units $1k$ iterations (by RMSprop optimizer, with a learning rate of 0.001 and a batch size of 128) with the data that are used to train the encoder and evaluate attribute classification accuracy on the validation dataset.

## 5.2 Results

**User anonymization:** Table 1 compares the user-anonymization performance. The value represents the lowest user-classification accuracy (the lower the better) with specific performance degradation compared to CNN on classification accuracy. For example, the columns with $0.01$ represent the lowest user-classification accuracy with less than $0.01$ point performance degradation. The best performance is underlined and highlighted in bold, and the second-best performance is only highlighted in bold. IIDM+ represents the variants that use the semantic alignment extension introduced in section 4. The results show the clear benefit of our proposal to inducing user invariance. Specifically, IIDM performs best on seven out of ten configurations. Note that the value with 'None' represents the method always reduce the label classification performance significantly.

**Domain generalization:** Table 2 summarizes the classification performance on two different datasets: MNISTR, and PACS. The top row of each table represents the test domain. We report

Table 2: Classification accuracies on unseen domains.

| | (a) MNISTR | | | | | | | (b) PACS | | | | |
|---|---|---|---|---|---|---|---|---|---|---|---|---|
| | M0 | M15 | M30 | M45 | M60 | M75 | Avg | photo | art | cartoon | sketch | Avg |
| **CNN** | 84.0± 1.7 | **99.1± 0.5** | 97.6± 0.9 | 91.9± 1.8 | 97.5± 0.5 | 87.7± 1.7 | 92.97 | 80.8± 1.3 | 58.1± 2.6 | 62.7± 2.6 | **60.6± 4.5** | 65.57 |
| **RevGrad** | 84.4± 1.6 | 98.8± 0.2 | 97.9± 0.8 | 92.1± 0.8 | 95.7± 2.2 | 85.9± 4.7 | 92.45 | 82.9± 1.3 | 57.2± 1.9 | 61.6± 0.6 | 54.6± 4.6 | 64.06 |
| **AII** | 83.8± 2.1 | 98.5± 0.4 | 97.4± 0.9 | 91.0± 1.4 | 97.0± 0.4 | 87.4± 2.4 | 92.52 | 81.1± 0.7 | 59.1± 1.7 | 60.7± 3.1 | **62.1± 3.0** | 65.75 |
| **AII+GP** | 86.2± 1.4 | 98.5± 0.2 | 97.9± 0.5 | 91.2± 0.7 | 97.0± 0.9 | **87.9± 2.0** | 93.11 | 81.8± 0.4 | 60.7± 0.2 | **64.0± 2.1** | 60.6± 3.3 | 66.76 |
| **CrossGrad** | 85.3± 0.9 | **98.9± 0.5** | 97.6± 0.8 | 90.9± 1.0 | **98.2± 0.4** | 87.5± 2.0 | 93.09 | 81.4± 1.8 | 58.1± 4.7 | 60.5± 3.1 | 60.5± 1.3 | 65.15 |
| **IIDM** | **88.0± 1.6** | 98.2± 1.0 | **98.1± 0.7** | **94.3± 0.8** | 98.0± 0.7 | **88.9± 1.3** | **94.25** | 82.9± 1.2 | **61.7± 1.5** | 63.4± 0.7 | 59.5± 0.5 | **66.89** |
| **IIDM+** | **88.3± 0.9** | 98.6± 0.5 | **98.1± 0.6** | 93.0± 1.8 | 98.1± 0.9 | 86.9± 2.5 | 93.85 | **84.8± 0.6** | **62.3± 1.6** | **64.8± 1.5** | 60.2± 2.5 | **68.04** |

(a) $\lambda$    (b) $\kappa$    (c) architecture of $\phi$

Figure 5: Comparison of AII and IIDM with different configurations on MNISTR dataset (M0 as test domain). The number in parenthesis represents the corresponding configuration.

the mean accuracy as well as the standard error of three different seeds (five seeds for MNISTR and three seeds for PACS). The best performance is underlined and highlighted in bold, and the second-best performance is only highlighted in bold. We can make the following observations. (1) IIDM and IIDM+ demonstrate the best or comparable performance on all conditions except sketch domain. Although the semantic alignment extension does not help the performance on a simpler task (MNISTR), it improves the performance on PACS dataset, giving approximately 1.0 point performance gain. (2) RevGrad and AII often fail to improve performance even when compared with a standard CNN. The score of AII+GP suggests that gradient penalty helps to improve the performance, but the improvements are lower than our proposal. (3) The Wilcoxon rank-sum test shows that IIDM is statistically better than CNN, RevGrad, AII, AII+GP, and CrossGrad with $p < 0.01$.

Figure 5 compares AII and IIDM on different (a) weighing parameter $\gamma$, (b) the number of the discriminator update $\kappa$, and (c) the network architecture of the discriminator. The dataset used is MNISTR with M0 as a test domain. In each figure, color represents a different method (red: AII, blue: IIDM) and marker denotes different configurations. The value represents the attribute classification accuracy (the lower the better invariant) by a post-hoc classifier $q_{eval}(a|z)$. For $\lambda$ we used 1.0 by default. For $\kappa$ and the architecture, we used default setting described in Section 5.1. The results show that our proposal is consistently to learn better invariant representations regardless of the choice of the hyperparameters. These results suggest that our proposal is better than searching such hyperparameters. Note that, $\lambda = 10.0$ for AII seems to attain better invariance, it was degenerated to the random representations and gives a random performance on the classification of $y$.

## 6    CONCLUSION

This paper presents a new method for invariance induction, called IIDM, by analyzing and extending current state-of-the-art adversarial invariance induction framework. This paper first examines the instability issue of AII both theoretically and empirically, indicating that AII has theoretical difficulty as it maximizes variational *upper* bound of the actual conditional entropy, and this fact leads AII to catastrophically fails even in simple cases. We then argue that a simple modification to AII can significantly stabilize the adversarial induction framework and achieve better invariant representations. The fundamental principle of our proposal is that a desirable invariance induction algorithm should also minimize the divergence between marginal distribution $p(z)$ between different attributes, as it is a requirement of maximum conditional entropy (Corollary 1) and missing in AII optimization. IIDM minimizes *discriminator matching* loss (Eq. 5), which is a proxy of the divergence between the marginals. On toy dataset, we compare our proposal with the adversarial invariance induction framework, and show that our proposal significantly stabilizes the optimization (Figure 3 and Figure 4). Two real-world tasks (user-anonymization in Table 1 and domain generalization in Table 1) also supports that our proposal achieve better invariance induction.

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

## A  PROOF OF THE PROPOSITION 1

*Proof.* Using the Lagrange multiplier method, the derivative of

$$L = -\sum_{a \in \mathcal{A}} p(a, z) \log p(a|z) + \lambda(1 - \sum_{a \in \mathcal{A}} p(a|z)) \tag{11}$$

is equal to zero for the maximum entropy $H(a|z)$. Solving the simultaneous equations, we can say $p(a_1|z) = p(a_2|z) = \cdots = p(a_K|z) = \frac{1}{K}$ for all $z \in \mathcal{Z}$ when the conditional entropy is maximized, and based on the definition, the conditional entropy becomes $-\log\frac{1}{K}$.

From Bayes' law and the uniform assumption of $p(a)$, $p(z|a_i) = p(z|a_j)$ holds $\forall a_i \neq a_j \in \mathcal{A}$ and $z \in \mathcal{Z}$. □

## B  RELATIONSHIPS BETWEEN DISCRIMINATOR MATCHING AND THE DIVERGENCE MINIMIZATION REQUIREMENTS

### B.1  PROOF OF THE EQ. 6

*Proof.*

$$
\begin{aligned}
D_{KL}(q_\phi^i(a)||q_\phi^j(a)) &= \sum q_\phi^i(a) \log q_\phi^i(a) - \sum q_\phi^i(a) \log \mathbb{E}_{p_\theta^j(z)}[q_\phi(a|z)] \\
&\leq \sum q_\phi^i(a) \log q_\phi^i(a) - \sum q_\phi^i(a)\mathbb{E}_{p_\theta^j(z)}[\log q_\phi(a|z)] \\
&= \mathbb{E}_{z_j \sim p_\theta^j(z)}\left[D_{KL}(q_\phi^i(a)||q_\phi(a|z_j))\right].
\end{aligned}
$$

□

The inequality shows that the discriminator matching objective is a variational upper bound of the KL-divergence between $q_\phi^i(a)$ and $q_\phi^j(a)$. Therefore, minimizing the proposed objective surely minimize the KL-divergence between $q_\phi^i(a)$ and $q_\phi^j(a)$. Note that, this inequality holds even when $q(a|z)$ is not optimal.

### B.2  DATA PROCESSING INEQUALITY OF THE F-DIVERGENCE

**Theorem 1.** *Consider a channel that produces $y$ given $x$ based on the $p(y|x)$. For any f-divergence* $D_f(p||q) = \mathbb{E}_q\left[f(\frac{p}{q})\right]$

$$D_f(p(y)||q(y)) \leq D_f(p(x)||q(x)) \tag{12}$$

*Proof.* Using $p(y|x)$, we consider the following joint distributions.

$$p(x, y) = p(y|x)p(x), \qquad q(x, y) = p(y|x)q(x),$$

The divergence between the tow joint distributions is

$$D_f(p(x,y)||q(x,y)) = \mathbb{E}_{q(x,y)}\left[f\left(\frac{p(y|x)p(x)}{p(y|x)q(x)}\right)\right] = D_f(p(x)||q(x)), \tag{13}$$

as $p(x, y)$ and $q(x, y)$ has common channel $p(y|x)$, and $p(y|x)$ is canceled. For other direction,

$$
\begin{aligned}
D_f(p(x,y)||q(x,y)) &= \mathbb{E}_{q(y)}\left[\mathbb{E}_{q(x|y)}\left[f\left(\frac{p(x,y)}{q(x|y)q(y)}\right)\right]\right] \\
&\geq \mathbb{E}_{q(y)}\left[f\left(\mathbb{E}_{q(x|y)}\left[\frac{p(x,y)}{q(x|y)q(y)}\right]\right)\right] \\
&= \mathbb{E}_{q(y)}\left[f\left(\frac{p(y)}{q(y)}\right)\right] = D_f(p(y)||q(y)).
\end{aligned}
$$

The inequality uses Jensen's inequality. Combining these two equations,

$$D_f(p(y)||q(y)) \leq D_f(p(x,y)||q(x,y)) = D_f(p(x)||q(x)). \tag{14}$$

□

As the KL divergence is also the family of the f-divergence, by replacing the $p(y|x)$ to $q_\phi(a|z)$, $p(x)$ to $p_\theta^i(z)$, $q(x)$ to $p_\theta^j(z)$, $p(y)$ to $q_\phi^i(a)$, and $p(y)$ to $q_\phi^j(a)$,

$$D_{KL}(p_\theta^i(z)||p_\theta^j(z)) \geq D_{KL}(q_\phi^i(a)||q_\phi^j(a)), \qquad (15)$$

which is equivalent to eq. 7.

It is noteworthily that if one can assume the common inverse mapping $p(x|y)$ for both joint distributions $p(x, y)$ and $q(x, y)$, then

$$D_f(p(y)||q(y)) = D_f(p(x)||q(x)). \qquad (16)$$

The proof is straightforward results with the above proof. In our case, $p(y|x)$ is parameterized by neural networks. In this case, if the $q_\phi(a|z)$ is invertible, then the inverse mapping exists. Intuitively speaking, the data process $q_\phi$ does not lose information, though the data is transformed to better explaining the attribute information (as $q_\phi$ is trained to discriminate the attribute).

