# OpenReview forum: "Stablizing Adversarial Invariance Induction by Discriminator Matching"
_ICLR.cc/2020/Conference — Reject_

### Official Review · AnonReviewer2 · 2019-10-22
**Official Blind Review #2**

**Rating:** 3

**Review:**

*Summary*

The paper proposes a new method to learn data-driven representations, being invariant to some specific nuisance factors which are detrimental for the selected (supervised) classification task.
Authors build upon the existing probabilistic framework termed Adversarial Invariant Induction (AII) from (Xie et al., 2017).

They claim to explore it under a both theoretical and practical point of view, demonstrating the limitations of maximizing a variational upper bound on conditional entropy as a proxy to achieve invariance.

Leveraging these observation, authors propose a novel method, called “invariance induction by discriminator matching” (IIDM) that is based on a regularized classification loss, penalized by a Kullbach-Leibler divergence between conditional distributions of the nuisance factor.

Extremely convincing experiments are carried out on a synthetic and a real benchmark in multi-source domain generalization (PACS).



*Pros*
1. The genesis of the proposed IIDM is extremely paced since smoothly derived from the AII framework.
2. Experimental results on a synthetic benchmark (a version of rotated MNIST) and on a popular benchmark for domain generalization (PACS) proved the effectiveness of IIDM



 *Cons*
1. The paper is hard to get, if the reader is not familiar with related literature
2. It is not fully clear from the paper which parts are original and which are inherited from prior work.
3. The structure of the paper needs to be improved (check my comments in the section beneath)
Some of the proposed methodologies are not clear (IIDM+)




*Detailed Comments*

The problem considered by authors is surely interesting and addressing a popular topic in computer vision and deep learning.

1. Unfortunately, the paper, as it is is hard to get for scholars which are not expert of the AII formalism, which, in my opinion is not enough detailed. Therefore, in my opinion clarity is something that authors should try to work hard on: for instance, during the rebuttal time, authors can write from scratch an entire new Section in which they explain in plain terms the main outcomes of their paper, without entering too much into technical details.


2. Additionally, the structure of the paper needs, in my opinion a major re-styling, still for the sake of better readability:
2.a. A visualization of the proposed method (for instance, using flow-diagrams) in comparison with the existing AII should help in rapidly getting the factors of novelty of the proposed IIDM method. I would also encourage authors to add a pseudo-code
2.b. Since authors claim two major contributions (understanding AII + IIDM), I would like to see those two contributions thoroughly presented and dissected in two separated sections of the paper. I am not fully convinced with the actual writing style in which the two contributions seem to be intertwined together.


3. Although already convincing, the experimental part can be improved:
3.a. Instead of a version of rotated-MNIST, authors can test on the “digits-five” setting (MNIST, MNIST-M, SVHN, UPS, SYN) as done in several works like http://openaccess.thecvf.com/content_cvpr_2018/papers/Xu_Deep_Cocktail_Network_CVPR_2018_paper.pdf.
3.b. In addition to multi-domain generalization, authors could also have tried more classical unsupervised domain adaptation settings or, even, single-source domain generalization as in https://papers.nips.cc/paper/7779-generalizing-to-unseen-domains-via-adversarial-data-augmentation.



*Final Evaluation*

I think that the main aspect that authors should face during the rebuttal is to make the paper more easy to read and better separate the two contributions (understanding AII and IIDM). What I am not convinced at all about the writing style of the authors since when reading the paper I am not always capable of understanding what is novel (since proposed by authors) and what is inherited from prior work. But, maybe, the reason for this is that I am not an expert of the specific related field - but, even so, I think that the paper needs to be understood from the broadest audience possible.

Instead, I am familiar with multi-source/single-source domain generalization (and adaptation) and, after my careful analysis of the experiments, I see a lot of potential in the approach. I would me more than interested in checking the performance of the proposed method over some of the novel benchmarks that I have recommended. It would be nice if authors add more experiments, but I know that this is always a complicated request during a rebuttal period.
Globally, if I were asked to only rate the experimental part, I would have promoted for full acceptance. Unfortunately, the theoretical part of the paper is not fully clear to me and, therefore, I am not confident in calling for a full acceptance only based on the experiments.

In brief, I would go for a weak reject, looking forward to the authors’ rebuttal and the opinion of the other Fellow Reviewers.

**Experience Assessment:**

I have read many papers in this area.

**Review Assessment: Checking Correctness Of Derivations And Theory:**

I did not assess the derivations or theory.

**Review Assessment: Checking Correctness Of Experiments:**

I assessed the sensibility of the experiments.

**Review Assessment: Thoroughness In Paper Reading:**

I read the paper at least twice and used my best judgement in assessing the paper.

---

> ### Author Response · Authors · 2019-11-12
> **Response to review #2**
>
> We thank the reviewer for detailed and encouraging comments. We have been updated the manuscript following the reviewers' comments. Please refer to the thread of "Summary of general updates.".
>
> Below are several clarifications. As the reviewer suggested, we mainly focus on improving the clarity in this rebuttal period.
>
> ---- Responses ----
> > The paper is hard to get, if the reader is not familiar with related literature
>
> We updated the manuscript to improve clarity. Please refer to the update 1, 2, 3 for more detail.
>
> > It is not fully clear from the paper which parts are original and which are inherited from prior work.
>
> We made several restructuring. Specifically,
> (1) We concentrate the explanation about AII in section 2 to make the paper (and the originality of this work) easy to understand. (2) Section 3 focuses on analyzing the practical issue of AII, which is the first contribution of this paper. (3) Section 4 derives a stable variant of AII, which is the second contribution of this paper.
>
> We also add a flow-diagrams and pseudo-code for a better explanation.
>
> > 3. Although already convincing, the experimental part can be improved: 3.a. Instead of a version of rotated-MNIST, authors can test on the “digits-five” setting (MNIST, MNIST-M, SVHN, UPS, SYN) as done in several works like http://openaccess.thecvf.com/content_cvpr_2018/papers/Xu_Deep_Cocktail_Network_CVPR_2018_paper.pdf. 3.b. In addition to multi-domain generalization, authors could also have tried more classical unsupervised domain adaptation settings or, even, single-source domain generalization as in https://papers.nips.cc/paper/7779-generalizing-to-unseen-domains-via-adversarial-data-augmentation.
>
> As suggested by the reviewer, we agree it would be interesting to see its performance on the other benchmarks. Unfortunately, we do not have much time and computational resources during this rebuttal period. Of course, we are happy to share the source code for additional tests after the acceptance of the paper.
> ----
>
> We look forward to hearing from you regarding our submission. We would be glad to respond to any further questions and comments that you may have.

---

### Official Review · AnonReviewer1 · 2019-10-24
**Official Blind Review #1**

**Rating:** 1

**Review:**

The paper points out that the practical behavior of AII assumes the optimality of the attribute classifier, which is rarely held in practice. And claims that the paper analyzes the practical behavior of AII both theoretically and empirically, indicating that AII has theoretical difficulty as it maximizes variational upper bound of the actual conditional entropy. Then it argues an ugly modification based on a wrong property of conditional entropy.

- The paper says that it analyzes AII theoretically and empirically. But it only shows the practical drawback of AII intuitively without any theoretical proof.
- In Section 3, the paper says : 'In general, maximizing the upper bound of the function of interest $f$ does not guarantee the minimizing the $f$ '.
- Also in the section 3, the paper says : 'Figure 2-(b) visualizes how distribution move during the optimization of AII on synthesized data'. And I want to ask why the caption of Figure 2-(b) is IIDM ?
- The proposition 1 on which the modification proposed in the paper is based will be not true when the distribution of attributes is not uniform by Bayses' s law which is used in the so called proof of the proposition 1. Which means that $p(z|a_i)=p(z|a_j) \Leftarrow p(a_i|z)=p(a_j|z)$ if and only if $p(a_i)=p(a_j)$.
- In the proof of equation 4 which is the main theorem of the total work. We can find $-\sum q_{\phi}^i(a)\log\mathbb{E}_{p_{\theta}^j(z)}[q_{\phi}(a|z)]\ge -\sum q_{\phi}^i(a)\mathbb{E}_{p_{\theta}^j(z)}[\log q_{\phi}(a|z)]$.  The inequality direction is reversed.


**Experience Assessment:**

I have published one or two papers in this area.

**Review Assessment: Checking Correctness Of Derivations And Theory:**

I assessed the sensibility of the derivations and theory.

**Review Assessment: Checking Correctness Of Experiments:**

I assessed the sensibility of the experiments.

**Review Assessment: Thoroughness In Paper Reading:**

I read the paper at least twice and used my best judgement in assessing the paper.

---

> ### Author Response · Authors · 2019-11-12
> **Response to review #1**
>
> We thank the reviewer for the detailed comments. We have been updated the manuscript following the reviewers' comments. Please refer to the thread of "Summary of general updates.".
>
> Below are several clarifications.
>
> --- Response ---
> > The paper says that it analyzes AII theoretically and empirically. But it only shows the practical drawback of AII intuitively without any theoretical proof.
>
> Section 3 of the new version focus on the theoretical analysis. If the reviewer feels it is too exaggerated to use the term "theoretical.", we could omit the word. We believe minor wording does not harm the overall contributions of this paper.
>
>  > In Section 3, the paper says : 'In general, maximizing the upper bound of the function of interest $f$ does not guarantee the minimizing the $f$'
>  > Also in the section 3, the paper says : 'Figure 2-(b) visualizes how distribution move during the optimization of AII on synthesized data'. And I want to ask why the caption of Figure 2-(b) is IIDM ?
>
> We are sorry that these are typographical mistakes. We fix the typo in the new version.
>
> > The proposition 1 on which the modification proposed in the paper is based
>
> The uniform $p(a)$ is an assumption of the analysis, as mentioned in the original manuscripts. We clarify this assumption more in the new version. Moreover, the analysis was indeed based on that assumption, but we believe it does not need to hold in practice. Our method works properly in the PACS dataset, where the attributes are imbalanced and still gives better performance than AII. We will add that to the discussion in the final manuscript if the reviewer thinks it is needed.
>
> > In the proof of equation 4 which is the main theorem of the total work.
>
> We admit our mistakes regarding equation 4. We fix the mistake in the new version. We believe that the revision does not harm the main contributions. Please refer to the update 5 in the thread of "Summary of general updates".
>
> ----
>
> We look forward to hearing from you regarding our submission. We would be glad to respond to any further questions and comments that you may have.

---

### Official Review · AnonReviewer3 · 2019-10-25
**Official Blind Review #3**

**Rating:** 3

**Review:**

** Summary
The paper studies the problem of representation learning under invariance constraints (i.e., the representation should be invariant wrt some attributes). The authors first review the adversarial invariance induction (AII) approach and they point out its limitations and then they propose a novel variant, which introduces an explicit regularization to minimize pairwise divergence (i.e., different attributes should lead to the same conditional distribution over the learned representation). The authors support the modified objective function both from a formal point of view and with an extensive empirical validation

** Evaluation
The paper lies a bit outside my area of expertise. Although the paper tries to capture intrinsic limitations of the AII approach and build a more stable algorithm, my impression is that too many elements in the discussion and derivation remain too vague at the current stage and they would require better and clearer explanation.

Detailed comments:
1- In many parts of the paper the notation is not very rigorous and sometimes it may create confusion. In general, the writing needs to be improved in many parts:
- In eq. 1, it is not explained what the expectation is wrt. It should be x drawn from p(x). But it would be better to make it explicit.
- The setting defined in sect.2 should be more rigorous: it is not clear what y is and what are the attributes we would like to be invariant for. This notation is not fully consistent with eq.1.
- In the first paragraph of Sect.3, it would be very helpful to have a more complete sketch of the algorithm. In general, you mention in the Sect. 2 that you focus on the supervised case, but then it is not clear whether this is the case across the paper.
2- While the intuition behind using the divergence is sound, as it is mentioned, it seems to suffer from the same issue as the original AII: minimizing a lower bound does not guarantee that we are minimizing the actual objective. Having the support of 0, does not seem to make it much more sensible.
3- As the description of the algorithm advances in Sect.4.2, it is clear that many additional choices need to be made in order to have a full workable algorithm (e.g., how to estimate q_\phi^i). In the paper, the actual algorithm is never reported in detail and this makes the experiments very hard to reproduce in my opinion.
4- At the best of my understanding, the algorithm may become more and more intractable as the number of attribute values grows. In fact, you need to check the divergence for each pair of values and estimated distributions.
5- The empirical analysis seems well executed and a good level of detail is reported on how the datasets are managed and the experiments are set up.

Minor comments:
- The caption of Fig.1 is not very clear. Many elements at this stage of the paper are not defined yet (e.g., "our proposal minimize the proxy of divergence ...").
- p2: "as the assumption of ... is rarely holds", remove "is"
- p3: "(encoder) that parameterized", remove "that"
- last paragraph of Sect.2 is very confusing.
- p4: "updation" is not a word
- p4: "In general, minimizing the upper bound..." does not seem correct.
- p4: "even such a simple case" -> "even in such a simple case"
- Sect 4.1 "the above theoretical", I would not really say there is much theory behind the analysis in the previous section.
- p5 "an ttribute" -> "an attribute"
- p6 "the average discriminator's perception" what is the perception?

**Experience Assessment:**

I do not know much about this area.

**Review Assessment: Checking Correctness Of Derivations And Theory:**

I assessed the sensibility of the derivations and theory.

**Review Assessment: Checking Correctness Of Experiments:**

I assessed the sensibility of the experiments.

**Review Assessment: Thoroughness In Paper Reading:**

I read the paper at least twice and used my best judgement in assessing the paper.

---

> ### Author Response · Authors · 2019-11-12
> **Response to review #3**
>
> We thank the reviewer for the detailed and encouraging comments. We have been updated the manuscript following the reviewers' comments. Please refer to the thread of "Summary of general updates.".
>
> Below are several clarifications.
>
> --- Response ---
> > In eq. 1, it is not explained what the expectation is wrt. It should be x drawn from p(x). But it would be better to make it explicit.
>
> The expectation is with respect to $p(x, a)$. We make it explicit in the new version.
>
> > The setting defined in sect.2 should be more rigorous: it is not clear what y is and what are the attributes we would like to be invariant for.
>
> We add several examples after the description of the problem settings. Please check the section 2 of the new version.
>
> > While the intuition behind using the divergence is sound, as it is mentioned, it seems to suffer from the same issue as the original AII: minimizing a lower bound does not guarantee that we are minimizing the actual objective. Having the support of 0, does not seem to make it much more sensible.
>
> We refine the explanation about the problem of AII (in section 3.2) and how it is alleviated in the proposed method (in section 4.1, the paragraph starting “Intuitively speaking”).  Please check the new version and the update 4 in the thread of "Summary of general updates".
>
> > As the description of the algorithm advances in Sect.4.2, it is clear that many additional choices need to be made in order to have a full workable algorithm (e.g., how to estimate q_\phi^i). In the paper, the actual algorithm is never reported in detail and this makes the experiments very hard to reproduce in my opinion.
>
> We refine the explanation about the proposed method. Please refer to the update of 1 in the thread of "Summary of general updates".
>
> > At the best of my understanding, the algorithm may become more and more intractable as the number of attribute values grows. In fact, you need to check the divergence for each pair of values and estimated distributions.
>
> We are afraid this is partially true but partially not true. More specifically, the exact computation of eq 5 (in the new version) becomes intractable as the number of attribute values grows. The naive implementation requires $O(K^2)$ computation, where $K$ is the number of attribute values. However, we introduce the moving centroids approximation to alleviate this intractable computation. Its order is $O(K)$ and can be trained with standard mini-batch training. We clarify this point in section 4.2 of the new version.
>
> ----
>
>
> We look forward to hearing from you regarding our submission. We would be glad to respond to any further questions and comments that you may have.

---

### Author Response · Authors · 2019-11-12
**Summary of general updates**

We thank all reviewers for their comments. They are insightful and help us to make our paper better. To address their comments, we substantially updated our paper to improve clarity. We admit several mistakes (including typos and a wrong inequality) and fixed the explanation accordingly. The revision does not harm the overall contributions of this paper.

Below is a summary of the major changes. We will clarify several points in the detailed response for each review.

[Update 1] Clarification of the proposed method (review #2 and #3):
We clarify our algorithm in multiple-level of granularity. In section 1 (Figure 1), we add flow-diagrams for helping the readers to get the concept of the proposed method rapidly. We also explicit a workable algorithm in the text and provide the pseud-code (in section 4.2).

[Update 2] Explanation about the previous method (review #2):
We concentrate the explanation about AII in section 2 to make the paper (and the originality of this work) easy to understand.

[Update 3] Organization of section 3 and section 4 (review #2):
We refine the explanation in section 3 and section 4 thoroughly. We move section 4.1 of the first version (explanation about the divergence minimization requirements) to section 3.2 in the new version. We also move simulation results about AII (the later parts of section 3 in the first version) to section 4.3 in the new version. We refine the text accordingly.

[Update 4] Explanation about the merit of the proposed method (review #3):
We refine the explanation about the problem of AII (in section 3.2) and how it is alleviated in the proposed method (in section 4.1, the paragraph starting “Intuitively speaking”).  Briefly, AII has an incentive to move the distribution far away even when the conditional entropy is successfully maximized (or equivalently, the marginal distributions of different attributes are aligned). Our proposed method stabilizes the behavior around this optimal point by restricting the update of the encoder to consider the location of the marginal distributions of a different attribute.

 [Update 5] Fix the inequality in section 4 (review #1):
We admit our mistake regarding the (part of) inequality regarding our discriminator matching loss and the divergence between the marginal distributions $p(z)$ of different attributes. We refine the explanation about the analysis of the relationships between the proposed method and the divergence minimization requirements (in section 4.1 and appendix B). Note that the revision *does not change* the conclusions. Specifically, discriminator matching loss is related to the required divergence minimization. The empirical validations support the merits of the proposal.

[Update 6] Typos (review #1 and #3):
We fix the typographical mistakes.


Thanks.

---

### Decision · Program_Chairs · 2019-12-19

**Decision:**

Reject

**Comment:**

The paper proposes a modification to improve adversarial invariance induction for learning representations under invariance constraints. The authors provide both a formal analysis and experimental evaluation of the method. The reviewers generally agree that the experimental evaluation is rigorous and above average, but the paper lacks clarity making it difficult to judge the significance of it. Therefore, I recommend rejection, but encourage the authors to improve the presentation and resubmit.